# Natural Variation and the Role of Zn_2_Cys_6_ Transcription Factors SdrA, WarA and WarB in Sorbic Acid Resistance of *Aspergillus niger*

**DOI:** 10.3390/microorganisms10020221

**Published:** 2022-01-20

**Authors:** Sjoerd J. Seekles, Jisca van Dam, Mark Arentshorst, Arthur F. J. Ram

**Affiliations:** 1TIFN, Agro Business Park 82, 6708 PW Wageningen, The Netherlands; s.j.seekles@biology.leidenuniv.nl; 2Department Molecular Microbiology and Biotechnology, Institute of Biology, Leiden University, Sylviusweg 72, 2333 BE Leiden, The Netherlands; j.l.van.dam@umail.leidenuniv.nl (J.v.D.); m.arentshorst@biology.leidenuniv.nl (M.A.)

**Keywords:** weak acid, screening, *Aspergillus niger*, transcription factor, food preservation, sorbic acid, strain diversity, CRISPR/Cas9

## Abstract

Weak acids, such as sorbic acid, are used as chemical food preservatives by the industry. Fungi overcome this weak-acid stress by inducing cellular responses mediated by transcription factors. In our research, a large-scale sorbic acid resistance screening was performed on 100 *A. niger sensu stricto* strains isolated from various sources to study strain variability in sorbic acid resistance. The minimal inhibitory concentration of undissociated (MIC_u_) sorbic acid at pH = 4 in the MEB of the *A. niger* strains varies between 4.0 mM and 7.0 mM, with the average out of 100 strains being 4.8 ± 0.8 mM, when scored after 28 days. MIC_u_ values were roughly 1 mM lower when tested in commercial ice tea. Genome sequencing of the most sorbic-acid-sensitive strain among the isolates revealed a premature stop codon inside the sorbic acid response regulator encoding gene *sdrA*. Repairing this missense mutation increased the sorbic acid resistance, showing that the sorbic-acid-sensitive phenotype of this strain is caused by the loss of SdrA function. To identify additional transcription factors involved in weak-acid resistance, a transcription factor knock-out library consisting of 240 *A. niger* deletion strains was screened. The screen identified a novel transcription factor, WarB, which contributes to the resistance against a broad range of weak acids, including sorbic acid. The roles of SdrA, WarA and WarB in weak-acid resistance, including sorbic acid, were compared by creating single, double and the triple knock-out strains. All three transcription factors were found to have an additive effect on the sorbic acid stress response.

## 1. Introduction

A significant portion of microbial food spoilage is caused by filamentous fungi, commonly referred to as moulds [1]. Several fungal species have the capacity to infect foods and beverages, and are able to proliferate in conditions with limited water availability, a lack of nitrogen, or after heat treatment [2]. Fungal spoilage can affect the visual appearance, taste and other properties of food products [3]. Additionally, the production of mycotoxins by food-spoiling fungi forms a direct risk for human health [4].

There are several ways in which the food industry preserves food and reduces microbial spoilage. Firstly, by the use of packaging, which prevents the access of microbes to food.; and secondly, by inactivation of the microorganisms in food by ionizing radiation and heat treatments, such as pasteurization and sterilization [5]. Another tactic involves growth inhibition of the microorganisms present on the foods; this includes storage at lowered temperatures, reducing the water activity of foods by drying products, or reducing oxygen availability by vacuum packaging. One other growth inhibition technique relies on the addition of chemical substances which reduce microbial growth, such as the addition of weak-acid preservatives [2,5].

Weak acids are food preservatives that cause growth inhibition on a broad spectrum of microorganisms. Weak acids are both fungistatic and bacteriostatic [6]. Commonly used weak acids in the food industry include sorbic acid, benzoic acid [7,8], propionic acid [9], lactic acid, acetic acid [10] and citric acid [11]. Sorbic acid can be added in its acid form, recognized by the European food additive number E200, but is more commonly added as the salt components sodium sorbate (E201), potassium sorbate (E202) or calcium sorbate (E203). Sorbic acid is added to food products such as condiments, bread, fruit jams, juices and soft drinks [12]. The concentration of weak acids allowed in foods and beverages is tightly controlled by governmental organizations such as the Food and Drug Administration (FDA) and the European Food and Safety Authority (EFSA). The maximum concentration of sorbates strongly depends on the food product, for example, the EFSA states that a maximum of 300 mg/L (2.67 mM) sorbate is allowed in flavored drinks (excluding dairy products), whereas a maximum of 500 mg/L (4.46 mM) sorbate is allowed in fruit and vegetable juices, and 2000 mg/L (17.84 mM) is allowed in processed cheeses [13]. The mode of action of these weak acids as preservatives is originally described in the “classical weak-acid theory” [14].

The classic weak-acid theory explains that weak acids, when present in a low pH, can cause the acidification of cells. In a liquid solution, a weak acid can be present in its undissociated form, in which the weak acid is present in its full molecule formation, or in its dissociated form in which the molecule has dissociated into the charged anion (WA^−^) and proton (H^+^). In solutions where the pH is equal to the pKa of the weak acid, the amount of undissociated acid (WAH) is equal to the amount dissociated acid (WA^−^ and H^+^). When the pH decreases, the proportion of undissociated acid (WAH) increases. Only weak acids in their undissociated form (WAH) are able to diffuse through the plasma membrane into the cytoplasm [15,16]. Therefore, when the pH < pKa, the weak-acid molecules diffuse through the plasma membrane and, because of the near neutral pH of the cytoplasm, the undissociated acids (WAH) are forced to dissociate into charged ions (WA^−^ and H^+^) [17,18]. The charged ions (WA^−^ and H^+^) are not able to diffuse back through the plasma membrane and accumulate in the cytoplasm, resulting in acidification of the cytoplasm [19]. The acidification of the cytoplasm is thought to inhibit glycolysis, since phosphofructokinase is sensitive to low pH [18]. In this classic theory, weak acids are thought to be most effective in solutions with a low pH.

However, recent studies have shown that the weak-acid theory is an oversimplistic view and alternative modes of action have been proposed. It has been shown for example that not all weak acids inhibit growth equally, and some weak acids do not even cause a lowered internal pH, indicating alternative mechanisms [20]. The toxicity of weak acids to microorganisms has also been linked to the lipophilic partition coefficient, suggesting that these compounds act on lipids or membranes [21]. A recent study in *Saccharomyces cerevisiae* showed that sorbic acid accumulates in mitochondrial membranes [20]. This study proposed that membrane-localized sorbic acid generates ROS that depletes mitochondrial respiratory function, through petite-cell formation and FeS cluster targeting. Similarly, in *Aspergillus niger* sorbic acid acts as a membrane-active compound which inhibits glucose and O_2_ uptake, thereby inhibiting conidial germination [22]. These examples show that weak-acid preservatives can have inhibiting effects on cells besides acidification of the cytoplasm, as described in the classical weak-acid theory.

Several food-spoiling fungi have been reported with resistance to these weak-acid preservatives, the most well-known species being spoilage yeasts. The spoilage yeast *Zygosaccharomyces bailii* has been reported to survive in up to 9.45 mM of sorbic acid and 11 mM of benzoic acid [3]. However, besides yeast species, mostly Ascomycetes are found as food contaminants [1]. Commonly found food-spoiling Ascomycetes include, for example, *A. niger*, *Paecilomyces variotii* and *Penicillium roqueforti* [2]. Therefore, preservative resistances of Ascomycetes have been investigated before, especially in relation to specific food products such as bread [23,24,25,26].

Several microbial resistance mechanisms protecting against weak-acid stress are known, and are most well-described for yeast species. Microbial resistance can be obtained by actively pumping out the weak acids, using, for example, ABC-type transporters such as Pdr12p from yeast [27]. Yeasts are also able to remodel plasma membranes and reinforce cell walls to decrease entry of weak acids through diffusion [28,29]. Additionally, in response to weak-acid stress, the energy threshold of yeast changes to prevent ATP depletion [29]. These cellular responses often require the activation of transcription factors, such as War1p, which mediates the expression of efflux pump Pdr12p [30].

Transcription factors play an important role in acquiring weak-acid resistance by food-spoiling fungi. *A. niger* has the ability to decarboxylate the weak acid’s sorbic acid and cinnamic acid, mediated by the enzymes phenyl transferase PadA and cinnamic acid decarboxylase CdcA [31,32]. These enzymes provide resistance towards sorbic acid and cinnamic acid, and are most effective during conidial germination and outgrowth [6]. CdcA and PadA are regulated by the Zn_2_Cys_6_-finger transcription factor sorbic acid decarboxylase regulator SdrA [32]. These genes are also present in other Aspergilli, which are able to grow on sorbic acid and cinnamic acid [32]. The deletion of *cdcA*, *padA*, or *sdrA* results in increased sensitivity towards sorbic acid and cinnamic acid, but the authors note that other factors might still be at play [32]. This suggests the presence of a separate yet uncharacterized set of genes that also add to the sorbic acid resistance of *A. niger*.

The transcription factor ‘weak-acid resistance A’ (WarA) has been recently described in *A. niger*, and is required for resistance against a range of weak-acid preservatives [33]. The knock-out strain (Δ*warA*) showed sensitivity to propionic, butanoic, pentanoic, hexanoic, sorbic and benzoic acids. WarA is described as having a CdcA-independent role in the sorbic acid resistance of *A. niger*, since the double knock-out strain lacking WarA and CdcA shows increased sensitivity to sorbic acid when compared to either of the single knock-out strains. Geoghegan and colleagues propose that WarA is possibly required for weak-acid resistance by regulating the expression of PdrA, an ATP binding cassette (ABC) type transporter, which the authors show is a homologue to Pdr12p, an ABC-type transporter known to pump out weak-acid anions in *S. cerevisiae* [27]. In *S. cerevisiae* Pdr12p is regulated by War1p, a Zn_2_Cys_6_-finger transcription factor, which binds to the weak-acid response element, WARE, in the Pdr12p promotor [30]. It should be noted that the *S. cerevisiae* War1p and *A. niger* WarA proteins are both Zn_2_Cys_6_ transcription factors, but do not show significant sequence similarity apart from the DNA binding domain [33].

In our research, a large-scale sorbic acid resistance screening was performed on 100 *A. niger* wild-type strains to study strain variability in sorbic acid resistance. Additionally, the screening of 240 transcription factor knock-out strains revealed the importance of multiple transcription factors in the weak-acid stress response, including WarB. We show that WarB is important for sorbic, benzoic, cinnamic, propionic and acetic acid stress resistance, and that the *warB* deletion has an additive effect on the sorbic acid resistance when combined with the *sdrA* and *warA* deletions.

## 2. Materials and Methods

### 2.1. Strains and Growth Conditions

All strains used in this study are listed in Table 1. The 100 *A. niger* wild-type strains were obtained from the CBS strain collection of the Westerdijk Institute of Fungal Biodiversity, Utrecht, the Netherlands. The *A. niger* strains were cultivated on malt extract agar (MEA, CM0059, Oxoid, Basingstroke, UK) plates for seven days at 30 °C, to harvest conidia for weak-acid stress resistance assays. Conidia were harvested by adding saline solution, consisting of 0.9% NaCl + 0.02% Tween 80 in demi water, to the plates, and gently scraping the spores with a sterile cotton swab; after this, the spore solution was filtered through a sterile filter (Amplitude EcoCloth, Contec, Vannes, France).

### 2.2. Sorbic Acid Sensitivity Screening by Liquid Assay

The sorbic acid (SA) minimal inhibitory concentration (MIC) of 100 *A. niger* strains was determined using a liquid assay utilizing 96-well plates based on previous research (van den Brule et al. unpublished results). In short, the 96-well plates contained malt extract broth (MEB, CM0057, Oxoid, Basingstroke, UK) and a concentration range of undissociated sorbic acid (0–9 mM in steps of 1 mM). MEB was adjusted to pH 4 by the addition of NaOH/HCl after autoclaving. The sorbic acid stock of 10 mM undissociated sorbic acid was made by dissolving 11.78 mM sorbic acid in warm MEB after autoclaving, subsequently adjusted to pH 4 with NaOH/HCl, and filter sterilized. The undissociated sorbic acid concentrations were calculated with the Henderson–Hasselbalch equation as defined by pH = pKa + log ([A^−^]/[HA]) [35]. Each well contained a total volume of 200 µL with a total of 10^4^ spores, by adding 10 µL of 10^6^ spores/mL spore stock, counted and diluted by using a TC20 automated cell counter (Bio-Rad, Hercules, CA, USA). Growth was scored after seven days and 28 days of growth at 25 °C using biological duplicates. The wells at the borders of the 96-well plates were used as water reservoirs and filled with 200 µL Milli-Q in order to prevent dehydration. To further limit dehydration and cross-contamination, the lids of the 96-well plates were kept closed during the experiment. Additionally, the 96-well plates were kept inside a closed box containing a falcon tube of water that functioned as an additional water reservoir, preventing dehydration of the wells.

The same assay was also performed to determine the MIC of 100 *A. niger* strains in commercial ice tea (Lipton Ice Tea Peach). Filter-sterilized and uncarbonated commercial ice tea was used. The pH of the ice tea was measured at 3.1 and not adjusted. A total of 10.2 mM sorbic acid was added to the ice tea, which was subsequently filter sterilized, resulting in a sorbic acid stock with a concentration of 10 mM undissociated sorbic acid.

### 2.3. Weak-Acid Sensitivity Screening by Spot Assays

Spot assays were performed on minimal medium plates (MM) containing 27.75 mM glucose and varying concentrations of weak acids. For the initial spot assay testing of 240 transcription factor knock-out strains, the following concentrations of (total) weak acids (pH = 4) were used: 4.5 mM sorbic acid, 2 mM and 3 mM benzoic acid, 2 mM and 3 mM cinnamic acid and 10 mM and 20 mM propionic acid. The minimal medium was prepared as described before [36], and set to pH 4 by the addition of NaOH/HCl after autoclaving. The weak acids tested were sorbic acid (Fluka chemika, Buchs, Switzerland), cinnamic acid (Fluka chemika, Buchs, Switzerland), benzoic acid (p-hydroxy-benzoic acid, Sigma Aldrich, St. Louis, MO, USA), propionic acid (Propionic acid sodium salt, Sigma Aldrich, St. Louis, MO, USA), lactic acid (Sigma Aldrich, St. Louis, MO, USA), citric acid (tri-sodium citrate dihydrate, VWR chemicals, Radnor, PA, USA) and acetic acid (acetic acid glacial, Biosolve chemicals, Valkenswaard, the Netherlands). Sorbic acid and cinnamic acid stock solutions were made in 70% ethanol. Acetic acid was used directly from the liquid stock solution. All other weak acids stocks were made in Milli-Q and filter sterilized before use.

Spot assays on MM containing weak acids were performed by spotting 5 µL of a 10^6^ conidia/mL spore stock solution, and the concentration determined by using a TC20 automated cell counter (Bio-Rad, Hercules, CA, USA), thereby inoculating a total of 5000 conidia per spot. Spot assay plates were cultivated for four days at 30 °C, after which pictures were taken and growth was determined, unless noted otherwise.

### 2.4. CRISPR/Cas9 Genome Editing in A. niger

Knock-out strains were constructed using a marker-free CRISPR/Cas9 genome editing approach as described previously [37]. All primers and plasmids used in this study are listed in Table 2 and Table 3, respectively. Single knock-out strains lacking the genes *sdrA* (An03g06580), *warA* (An08g08340) and *warB* (An11g10870) were made in MA234.1. Additionally, all possible combinations of knock-out strains were made; (Δ*sdrA*, Δ*warA*), (Δ*warA*, Δ*warB*), (Δ*sdrA*, Δ*warB*) and (Δ*sdrA*, Δ*warA*, Δ*warB*) (Table 1).

A schematic overview of the technique used for complementation to replace *sdrA* in wild-type strain CBS 147320 by the *sdrA* locus obtained from laboratory strain N402 is shown in Appendix A. CRISPR/Cas9 plasmid pMA433.2 was used to create a double strand break in *sdrA*. The target sequence was designed using CHOPCHOP predictors [39]. The repair DNA was constructed by amplifying the gene, as present in N402, by PCR. The donor DNA contained two newly introduced silent point mutations in order to eliminate further Cas9 endonuclease activity after a homology-directed repair event has taken place. In short, the reverse primer of the 5′ part of the gene (p2r sjs28) and the forward primer of the 3′ part of the gene (p3f sjs28) were designed to contain an overlapping sequence. This overlapping sequence contained the two newly introduced silent point mutations. The repair DNA was subsequently constructed by fusion PCR. PCR reactions to obtain repair DNA for the complementation were performed using Phusion™ High-Fidelity DNA Polymerase (Thermo scientific, Waltham, USA) with its appropriate buffer, and protocol as prescribed by the manufacturer. Transformation was performed using a PEG-mediated protocol described previously [36], with few exceptions. Protoplast formation of the wild-type *A. niger* strain CBS 147320 was seen after 2.5 h of incubating. After five days of growth, transformants were single streaked on MM + hygromycin (100 µg/mL) for purification, and afterwards on MM, and MM + hygromycin, for the subsequent removal of selection pressure to select for transformants that lost the CRISPR/Cas9-containing plasmid. Transformants were purified and tested for sorbic acid resistance. Four out of 55 transformants were found to have increased sorbic-acid resistance, and sequencing of the *sdrA* locus of those mutants confirmed integration of the donor DNA at the *srdA* locus.

## 3. Results

### 3.1. Natural Variation of Sorbic Acid Resistance among 100 A. niger Sensu Stricto Strains

All 100 wild-type *A. niger sensu stricto* strains were obtained from the CBS collection, Westerdijk Fungal Biodiversity Institute, Utrecht, the Netherlands. These strains originate from all over the word and were isolated from diverse sources (Table 1). In order to investigate the food-spoiling capacity of these strains, a 96-well plate assay was performed testing sorbic acid resistance of *A. niger* strains in two types of liquid media: malt extract broth (MEB) (Figure 1) and commercial ice tea (Lipton Peach Ice Tea) (Figure 2). The 100 *A. niger* strains were subjected to 0–9 mM of undissociated sorbic acid and scored for growth after seven and 28 days.

The average minimal inhibitory concentration of undissociated acid (MIC_u_) of the 100 *A. niger* strains was determined in both MEB and a commercial ice tea, and shown together with the MIC_u_ of the most resistant and sensitive strain in Table 4. The fungal static effect of sorbic acid became apparent by determining the MIC after prolonged incubation (28 days) compared to seven days. When scoring after 28 days instead of seven days, the MIC increased 1.1 mM and 0.9 mM in MEB and ice tea, respectively. The average MIC_u_ of sorbic acid when tested in commercial ice tea was roughly 1 mM lower than the values obtained in MEB (Table 4).

### 3.2. Genome Sequencing and SNP Analysis of the Most Sorbic-Acid-Sensitive Strain CBS147320

The genome of strain CBS 147320 has been previously sequenced (BioProject ID PRJNA743902). When analysing the genome of this strain, we discovered an SNP inside the *sdrA* gene (G1296A, located in the Fungal-specific transcription factor domain PF04082), resulting in a premature stop codon. Therefore, the SdrA protein (originally 657 amino acids long) is truncated in the sorbic-acid-sensitive wild-type strain CBS 147320, and is only 384 amino acids long. In order to test whether the nonsense mutation in *sdrA* is responsible for the high sensitivity towards sorbic acid, a complementation study was designed to restore the mutation resulting in a stop codon (TGA) back to the codon (TGG) found in the wild-type *sdrA* gene, which is found in other isolates and the N402 strain (for complementation methodology see Appendix A). In short, fungal transformations were performed, in which a double-strand break (DSB) was introduced in the *sdrA* locus in CBS 147320 using CRISPR/Cas9 and a *sdrA* specific gRNA. The donor DNA containing a truncated copy of *sdrA* amplified from the lab strain N402 (containing the wild-type gene of *sdrA*) was provided during the transformation. Transformants were obtained, and correct restoration of the point mutation in CBS 147320 was confirmed in four transformants by diagnostic PCR and subsequent sequencing of the PCR product. The four transformants were analysed for their sorbic acid resistance by performing a spot-assay on MM plates containing 1, 2, 3 and 4 mM sorbic acid at pH = 4 (Figure 3). The parental strain CBS 147320, a *sdrA* deletion strain and sorbic-acid-resistant strain CBS 113.50 were taken along as controls. The four transformants show increased sorbic acid resistance compared to the parental strain CBS 147320, having visible growth after four days on MM plates containing 3 mM sorbic acid. These results indicate that the weak-acid sensitivity of the wild-type strain CBS 147320 could be restored by introducing the wild-type *sdrA* gene back into the genome.

### 3.3. Screening for Transcription Factors That Are Related to Weak-Acid Stress Resistance

In order to identify additional transcription factors involved in weak-acid stress resistance of *A. niger*, a library containing 240 *A. niger* transcription factor knock-out strains was screened for sorbic acid, cinnamic acid, benzoic acid and propionic acid resistance (knock-out library made by Arentshorst, van Peij, Pel and Ram, unpublished data). A selection of transcription factor knock-out strains with interesting phenotypes were reevaluated using smaller concentration steps (Figure 4).

The screening of 240 transcription factor knock-out strains revealed multiple candidate transcription factors involved in weak-acid stress resistance in *A. niger*, including the *atfA* homologue putatively involved in the general stress response [40], the *nsdC* homologue, the *hapX* homologue, the *acuK* homologue and *creA*; the latter of these is the main regulator of carbon catabolite repression [41]. Additionally, knock-out strains lacking transcription factors An12g08510 and An11g10870, with no clear homologues, showed reduced growth on plates containing weak acids. The knock-out strain lacking gene An11g10870 was specifically interesting, showing a severe growth reduction on four out of the five weak acids tested: benzoic acid, propionic acid, benzoic acid and acetic acid. This gene was studied further and named WarB for ‘weak-acid resistance B’. WarB is a Zn_2_Cys_6_ transcription factor consisting of only 307 amino acids and is significantly shorter than the other sorbic acid response regulators SdrA (628 amino acids) and WarA (777 amino acids). No clear homology exists between WarB and the previously described transcription factors WarA, SdrA, or the sorbic acid response regulator in yeast, War1p [31,33]. In order to further investigate the effects of the *warB* deletion, single and combination knock-out strains lacking *sdrA*, *warA* and *warB* were made using CRISPR/Cas9 genome editing. The proper deletion of the strain was verified by diagnostic PCR (Appendix A) and these strains were tested for their resistance against weak-acid preservatives, using a spot assay (Figure 5).

The single knock-out strain Δ*warB* was more sensitive towards sorbic acid, benzoic acid and cinnamic acid compared to its parental strain. The Δ*warB* single knock-out strain was more sensitive towards benzoic acid when compared to the Δ*sdrA* or Δ*warA* single knock-out strains. The double knock-out strain Δ*sdrA*, Δ*warB* showed severely reduced growth in the presence of cinnamic acid when compared to either Δ*sdrA* or Δ*warB* single knock-out strains. All three transcription factors, SdrA and WarA and WarB, were involved in sorbic acid resistance, as shown by the higher sensitivity of the triple knock-out strain than of any of the double knock-out strains, indicating that these three transcription factors work side-by-side to generate the regular sorbic acid stress response. Additionally, the sorbic acid sensitivity of the knock-out strains was investigated in a liquid assay (MEB) in 96-well plates, similar to the experiment performed on wild types in Figure 1, confirming the same impact of SdrA, WarA and WarB on sorbic acid resistance in liquid (Appendix A).

## 4. Discussion

Heterogeneity among different (natural) isolates of the same species in relation to the weak-acid stress resistance of fungi has been reported before. For example, in the case of spoilage yeast *Z. bailii* [3], variation in MIC values among strains was reported to vary between 4.5 mM and 9.5 mM (mean 7.1 mM). In our study, the sorbic acid MIC_u_ of *A. niger* was determined for 100 strains, and showed an average of 4.8 ± 0.8 mM in MEB and 3.8 ± 0.5 mM in commercial ice tea when scored after 28 days (Table 4). These findings for *A. niger* are consistent with earlier reports. A recent study tested three *A. niger* strains and one *Aspergillus tubingensis* strain and reported sorbic acid MIC_u_ values of these strains between 2.88 Mm–4.80 mM [23]. Therefore, most *A. niger* strains will survive the maximum allowed sorbates in flavored drinks (2.67 mM), some will survive the limits allowed in fruit juices (4.46 mM) and no strains will survive the limits allowed in processed cheeses (17.84 mM) [13]. However, it is important to note the outliers, specifically the most sorbic—acid-resistant *A. niger* strain out of the 100 reported in our study (Table 4), CBS 113.50, with a sorbic acid MIC_u_ value of 7 mM. This means that, depending on the strains found in any specific food processing facility, one might need 7 mM of undissociated sorbic acid to reliably prevent growth of *A. niger*. Additionally, MIC_u_ values are dependent on the medium used, as strains consistently showed lower MIC_u_ values in commercial ice tea when compared to relatively rich medium MEB. We noticed that sporulation was limited in the ice tea medium, and the 96-well plates showing growth were not as densely packed with mycelium when compared to the same assay performed in MEB. Perhaps ice tea is a relatively poor growth medium for *A. niger*, thereby lowering the minimal concentration of sorbic acid needed to prevent outgrowth. No clear relationship between an isolated source and sorbic acid resistance was found. *A. niger* strains isolated as food contaminants were not the most sorbic-acid-resistant strains in our study. Only two strains, CBS 113.50 and DTO 146-E8, belong consistently to the top five most sorbic-acid-resistant strains in both MEB and commercial ice tea, and these strains were isolated from leather and an indoor environment, respectively (Table 1). The most sorbic-acid-sensitive strain in both MEB and commercial ice tea, CBS 147320, was isolated from a grape.

The most sorbic-acid-sensitive strain, CBS 147320, had an SNP inside the *sdrA* gene, resulting in a premature stop codon. The sorbic acid resistance could be increased again by replacing the SNP with the ‘normal’ base present in all other *A. niger* strains (Figure 3). Therefore, the most sorbic-acid-sensitive wild-type strain found, originally isolated from a grape in Australia, was in fact a strain lacking activity of the transcription factor SdrA. This indicates that the transcription factors involved in the sorbic acid response are important, not solely for our understanding of the molecular mechanisms behind fungal sorbic acid resistance, but also as an important factor within the observed strain variability of *A. niger*.

A spot-assay testing weak-acid resistance of 240 *A. niger* knock-out strains, each lacking a single transcription factor, revealed transcription factors that are potentially involved in the weak-acid stress response. One transcription factor is a homologue of the general stress response regulator AtfA [40]. Three transcription factors which, upon deletion, reduced the resistance towards sorbic acid are homologues of genes regulating siderophores and iron uptake, HapX, NsdC and AcuK [42,43,44]. AcuK is known to be essential for growth on gluconeogenic carbon sources, and its reduced resistance could possibly be caused by a metabolic imbalance and reduced catabolism of sorbic acid. However, AcuK is also required for iron uptake in *Aspergillus fumigatus* and regulates a set of genes involved in iron homeostasis, including the gene *hapX* [42]. Recently, researchers have shown that NsdC regulates many genes in *A. fumigatus* and impacts stress resistance against cell wall damaging agents; however, NsdC also regulates the expression of siderophores and genes involved in iron uptake, again, including *hapX* [44]. Transcription factor HapX is best known for its role in iron homeostasis; however, researchers have shown that the HapX protein also has a putative role in mitochondrial metabolism in *A. fumigatus*, more than 30% of the target genes of HapX have a function in the mitochondria [45]. As discussed in the introduction, recent publications have disputed the classical weak-acid theory of cytosolic acidification, and instead propose that weak acids disrupt mitochondrial respiration [20]. It is interesting to note that the effectivity of weak acids in the mitochondrial membrane could perhaps explain why the *acuK*, *nsdC* and *hapX* knock-out strains were linked to weak-acid stress sensitivity. The identification of the transcription factors involved in mitochondrial function and iron uptake is in line with previous screening of the yeast deletion library [46], and suggests the importance of maintaining a reducing intracellular environment for the sorbic acid resistance of *A. niger*. The Δ*creA* strain also seems impacted by weak acids. Since the Δ*creA* strain is impacted in carbon catabolite repression, the strain does not limit itself to glucose uptake and metabolism. Perhaps the active uptake of weak acids as a potential carbon source in the Δ*creA* strain is the cause of its weak-acid sensitive phenotype.

Another interesting gene was the putative transcription factor An11g10870, dubbed WarB. We analysed the available expression data of *A. niger* growing in the presence of sorbic acid, in order to investigate the expression of the *warB* gene during sorbic acid stress [33]. The *warB* gene shows induction (logFC = 5.5) in the presence of sorbic acid compared to the control, indicating the possible involvement of WarB in the sorbic acid stress response. The deletion of *warB* resulted in increased sensitivity towards benzoic, sorbic, cinnamic, propionic and acetic acid (Figure 3 and Figure 4). Double and triple knock-out strains, which were lacking *sdrA*, *warA* and/or *warB*, were made in *A. niger*, to investigate the relative importance of each transcription factor in the weak-acid stress response. All three transcription factors seem to contribute to the sorbic acid resistance of *A. niger*, as indicated by the high sorbic acid sensitivity of the triple knock-out strain. Interestingly, the Δ*warA*, Δ*warB* double knock-out strain seems to be slightly more resistant to both sorbic acid and cinnamic acid than the Δ*warB* single knock-out strain. Perhaps this finding indicates a compensatory effect, where *sdrA* is upregulated in the absence of *warA* and *warB*; however, further research is needed to confirm this hypothesis.

It is currently not known which genes’ transcription factor WarB regulates. However, a set of 18 genes show a significant positive correlation (Spearman coefficient > 0.5) in co-expression with transcription factor *warB*, when analysing co-expression networks of *A. niger* on FungiDB [47] (Appendix A). Interestingly, one of these 18 genes, An18g01150, codes for a putative fluconazole transporter, a homologue to FLU1 from *Candida albicans* [48] (48% identity, 85% coverage). Recent reports have suggested that the deletion of the MFS-type transporter FLU1 in *C. albicans* has no major impact on fluconazole resistance, but rather makes *C. albicans* susceptible to both histatin 5 [49] and the weak acid mycophenolic acid [48,50]. Another significantly co-expressed gene, An02g04160, codes for a putative mitochondrial phosphate translocator (MPT) which plays a crucial role in mitochondrial respiration, similar to genes regulated by the previously mentioned transcription factors AcuK, NsdC and HapX. Future research could focus on the target genes regulated by WarB and the role of mitochondrial respiration in weak-acid stress, thereby expanding our knowledge on the weak-acid stress response of filamentous fungi including *A. niger*.

## Figures and Tables

**Figure 1 microorganisms-10-00221-f001:**
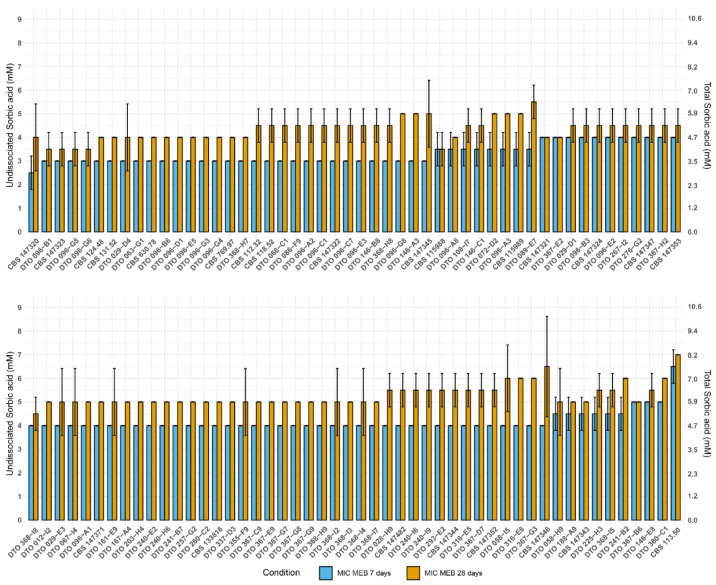
MIC assay showing strain diversity of 100 *A. niger* strains grown in MEB. Mean MIC values for each strain was determined after seven days (blue), and 28 days (orange) of growth at 25 °C, from biological duplicates. The error bar indicates the standard deviation between the duplicates. The primary Y-axis indicates the undissociated sorbic acid concentration, whereas the secondary Y-axis indicates the total sorbic acid concentration.

**Figure 2 microorganisms-10-00221-f002:**
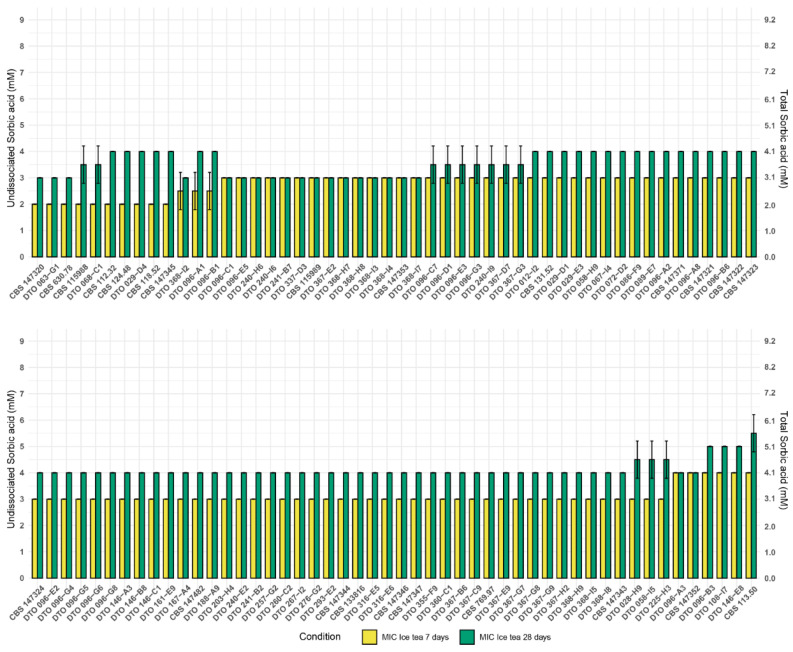
MIC assay showing strain diversity of 100 *A. niger* strains grown in ice tea. Mean MIC values for each strain was determined after seven days (yellow), and 28 days (green) of growth at 25 °C, from biological duplicates. The error bar indicates the standard deviation between the duplicates. The primary Y-axis indicates the undissociated sorbic acid concentration, whereas the secondary Y-axis indicates the total sorbic acid concentration.

**Figure 3 microorganisms-10-00221-f003:**
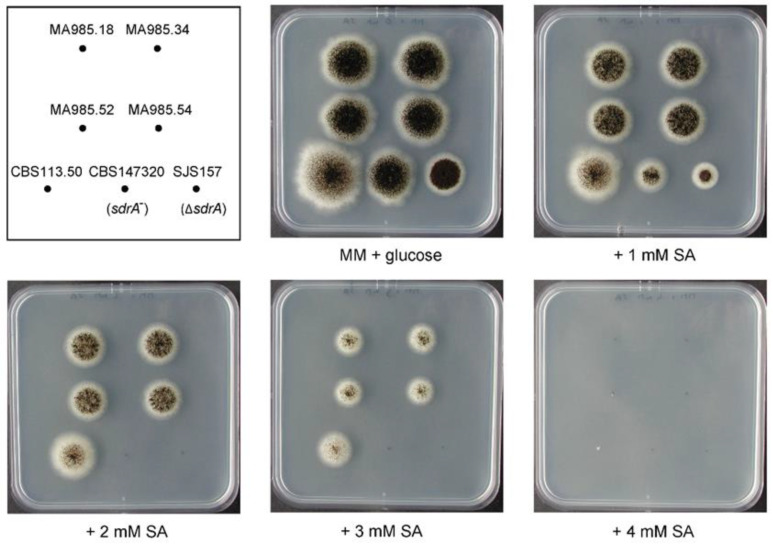
Phenotypic screen of CBS 147320 transformants with a restored *sdrA* locus on sorbic acid. The transformants with a confirmed restored *sdrA* locus were tested on sorbic acid resistance. The spot assay was conducted on MM + glucose with the addition of sorbic acid (SA), concentrations given of undissociated sorbic acid. Conidia were spotted and plates were subsequently grown for four days at 30 °C. Several controls were taken along; they are sorbic-acid-resistant strain CBS 113.50, sorbic-acid-sensitive wild-type strain CBS 147320 (*sdrA*^−^) and sorbic-acid-sensitive knock-out strain SJS157 (Δ*sdrA*). Transformants were resistant to SA when compared to the parental strain CBS 147320 and the Δ*sdrA* deletion strain SJS157. The four transformants with a restored *sdrA* gene show SA resistance comparable to the most resistant wild-type strain CBS 113.50.

**Figure 4 microorganisms-10-00221-f004:**
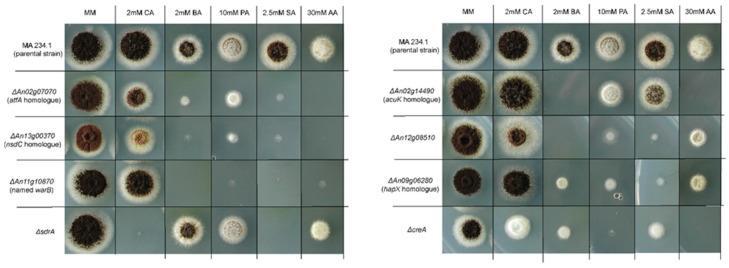
Transcription factor knock-out strains sensitive to weak-acid stress. The spot assay was conducted on MM containing glucose (pH = 4). Weak-acid concentrations listed are total weak-acid concentrations added, the undissociated acid concentrations (pH = 4) are 1.5 mM cinnamic acid (CA), 1.2 mM benzoic acid (BA), 8.8 mM propionic acid (PA), 2.1 mM sorbic acid (SA) and 25.6 mM acetic acid (AA). Growth was scored and pictures were taken after five days of incubation at 30 °C. The top row contains parental strain MA 234.1 (Δ*kusA*) two times the same identical spots, as a control for growth comparison.

**Figure 5 microorganisms-10-00221-f005:**
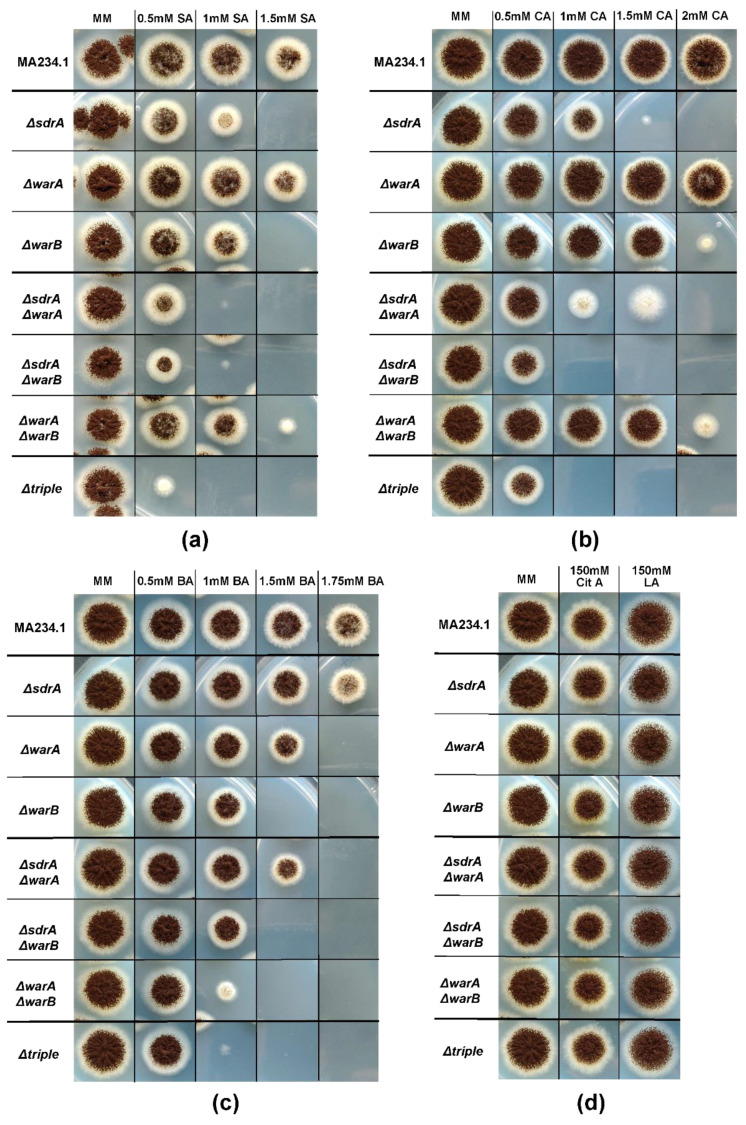
Weak-acid stress resistance of knock-out strains lacking *sdrA*, *warA* and/or *warB*. Conidia are spotted on MM containing glucose and weak acid, pH 4, grown for four days at 30 °C. Growth was compared to the growth phenotype of the parental strain MA234.1 (Δ*kusA*). All concentrations of weak acids added listed are total weak-acid concentrations: (**a**) sorbic acid (SA) stress resistance. The triple knock-out strain is the most sensitive for sorbic acid stress. A double knock-out strain Δ*warA*, Δ*warB* is less sensitive than the single knock-out strain Δ*warB*; (**b**) cinnamic acid (CA) stress resistance. The Δ*sdrA*, Δ*warB* strain is the most sensitive to cinnamic acid, the *warA* deletion does not seem to affect cinnamic acid stress resistance; (**c**) benzoic acid (BA) stress resistance. The *warB* deletion has the largest effect on the benzoic acid resistance; (**d**) citric acid (Cit A) and lactic acid (LA) stress resistance seemed unaffected.

**Table 1 microorganisms-10-00221-t001:** Strains used in this study.

CBS Number Strain	DTO Number Strain	Genotype	Parental Strain	Isolated from	Species	Obtained from
CBS 113.50	DTO 008-C3	wild type	-	Leather	*Aspergillus niger*	Westerdijk Fungal Biodiversity Institute
CBS 554.65	DTO 012-I2	wild type	-	Tannic–gallic acid fermentation, Conneticut, USA	*Aspergillus niger*	Westerdijk Fungal Biodiversity Institute
CBS 110.30	DTO 028-H9	wild type	-	Göttingen, Germany	*Aspergillus niger*	Westerdijk Fungal Biodiversity Institute
CBS 112.32	DTO 028-I3	wild type	-	Japan	*Aspergillus niger*	Westerdijk Fungal Biodiversity Institute
CBS 124.48	DTO 029-B1	wild type	-	Unknown	*Aspergillus niger*	Westerdijk Fungal Biodiversity Institute
CBS 131.52	DTO 029-C3	wild type	-	Leather	*Aspergillus niger*	Westerdijk Fungal Biodiversity Institute
CBS 263.65	DTO 029-D1	wild type	-	Copenhagen, Denmark	*Aspergillus niger*	Westerdijk Fungal Biodiversity Institute
CBS 103.66	DTO 029-D4	wild type	-	Unknown	*Aspergillus niger*	Westerdijk Fungal Biodiversity Institute
CBS 623.78	DTO 029-E3	wild type	-	France	*Aspergillus niger*	Westerdijk Fungal Biodiversity Institute
CBS 117.52	DTO 058-H9	wild type	-	Unknown	*Aspergillus niger*	Westerdijk Fungal Biodiversity Institute
CBS 118.52	DTO 058-I1	wild type	-	Unknown	*Aspergillus niger*	Westerdijk Fungal Biodiversity Institute
CBS 139.52	DTO 058-I5	wild type	-	Kuro-koji, Japan	*Aspergillus niger*	Westerdijk Fungal Biodiversity Institute
CBS 115988	DTO 059-C7	wild type	-	Unknown	*Aspergillus niger*	Westerdijk Fungal Biodiversity Institute
CBS 123906	DTO 063-G1	wild type	-	Ryuku, Japan	*Aspergillus niger*	Westerdijk Fungal Biodiversity Institute
CBS 630.78	DTO 067-H7	wild type	-	Army equipment, South Pacific Islands	*Aspergillus niger*	Westerdijk Fungal Biodiversity Institute
CBS 118.36	DTO 067-I4	wild type	-	Chemical, USA	*Aspergillus niger*	Westerdijk Fungal Biodiversity Institute
CBS 126.49	DTO 068-C1	wild type	-	Unknown	*Aspergillus niger*	Westerdijk Fungal Biodiversity Institute
-	DTO 072-D2	wild type	-	Indoor air of archive, the Netherlands	*Aspergillus niger*	Westerdijk Fungal Biodiversity Institute
-	DTO 086-F9	wild type	-	Filter flow cabinet, Westerdijk institute, Utrecht, the Netherlands	*Aspergillus niger*	Westerdijk Fungal Biodiversity Institute
-	DTO 089-E7	wild type	-	Air in crawling space, Eindhoven, the Netherlands	*Aspergillus niger*	Westerdijk Fungal Biodiversity Institute
-	DTO 096-A1	wild type	-	Wall down in the Lechuguilla Cave, Carlsbad, New Mexico, USA	*Aspergillus niger*	Westerdijk Fungal Biodiversity Institute
-	DTO 096-A2	wild type	-	Soil from dirt road, Isla Santa Cruz, Galapagos islands, Ecuador	*Aspergillus niger*	Westerdijk Fungal Biodiversity Institute
-	DTO 096-A3	wild type	-	Spent coffee (mouldy growth), Denmark	*Aspergillus niger*	Westerdijk Fungal Biodiversity Institute
CBS 147371	DTO 096-A5	wild type	-	Green coffee bean, Coffee Research Station, Netrakonda, India	*Aspergillus niger*	Westerdijk Fungal Biodiversity Institute
CBS 147320	DTO 096-A7	wild type	-	Grape, Australia	*Aspergillus niger*	Westerdijk Fungal Biodiversity Institute
-	DTO 096-A8	wild type	-	Artic soil, Svalbard, Norway	*Aspergillus niger*	Westerdijk Fungal Biodiversity Institute
CBS 147321	DTO 096-A9	wild type	-	Artic soil, Svalbard, Norway	*Aspergillus niger*	Westerdijk Fungal Biodiversity Institute
-	DTO 096-B1	wild type	-	Rice starch, imported to Denmark	*Aspergillus niger*	Westerdijk Fungal Biodiversity Institute
-	DTO 096-B3	wild type	-	Pepper, imported to Denmark	*Aspergillus niger*	Westerdijk Fungal Biodiversity Institute
-	DTO 096-B6	wild type	-	Saffron powder, from Kenya imported to Denmark	*Aspergillus niger*	Westerdijk Fungal Biodiversity Institute
-	DTO 096-C1	wild type	-	Unknown	*Aspergillus niger*	Westerdijk Fungal Biodiversity Institute
CBS 147322	DTO 096-C6	wild type	-	Coffee, Brazil	*Aspergillus niger*	Westerdijk Fungal Biodiversity Institute
-	DTO 096-C7	wild type	-	Unknown	*Aspergillus niger*	Westerdijk Fungal Biodiversity Institute
-	DTO 096-D1	wild type	-	Unknown	*Aspergillus niger*	Westerdijk Fungal Biodiversity Institute
CBS 147323	DTO 096-D7	wild type	-	Raisin, Fabula, Turkey	*Aspergillus niger*	Westerdijk Fungal Biodiversity Institute
CBS 147324	DTO 096-E1	wild type	-	Unknown	*Aspergillus niger*	Westerdijk Fungal Biodiversity Institute
-	DTO 096-E2	wild type	-	Unknown	*Aspergillus niger*	Westerdijk Fungal Biodiversity Institute
-	DTO 096-E3	wild type	-	Unknown	*Aspergillus niger*	Westerdijk Fungal Biodiversity Institute
-	DTO 096-E5	wild type	-	Unknown	*Aspergillus niger*	Westerdijk Fungal Biodiversity Institute
CBS 101700	DTO 096-G3	wild type	-	Japan	*Aspergillus niger*	Westerdijk Fungal Biodiversity Institute
CBS 101706	DTO 096-G4	wild type	-	Soy bean	*Aspergillus niger*	Westerdijk Fungal Biodiversity Institute
CBS 101707	DTO 096-G5	wild type	-	Broiler mixed feed	*Aspergillus niger*	Westerdijk Fungal Biodiversity Institute
CBS 101708	DTO 096-G6	wild type	-	Uknown	*Aspergillus niger*	Westerdijk Fungal Biodiversity Institute
CBS 121047	DTO 096-G8	wild type	-	Coffee bean, Thailand	*Aspergillus niger*	Westerdijk Fungal Biodiversity Institute
-	DTO 108-I7	wild type	-	Indoor environment, Thailand	*Aspergillus niger*	Westerdijk Fungal Biodiversity Institute
CBS 120.49	DTO 146-A3	wild type	-	USA	*Aspergillus niger*	Westerdijk Fungal Biodiversity Institute
CBS 101698	DTO 146-B8	wild type	-	Mesocarp finga-coffee bean, Kenya	*Aspergillus niger*	Westerdijk Fungal Biodiversity Institute
CBS 101705	DTO 146-C1	wild type	-	Carpet dust from school, Canada	*Aspergillus niger*	Westerdijk Fungal Biodiversity Institute
-	DTO 146-E8	wild type	-	Indoor environment, Hungary	*Aspergillus niger*	Westerdijk Fungal Biodiversity Institute
-	DTO 161-E9	wild type	-	Bamboo sample, Ho Chi Minh city, Vietnam	*Aspergillus niger*	Westerdijk Fungal Biodiversity Institute
-	DTO 167-A4	wild type	-	Margarine, Belgium	*Aspergillus niger*	Westerdijk Fungal Biodiversity Institute
CBS 147482	DTO 175-I5	wild type	-	Surface water, Portugal	*Aspergillus niger*	Westerdijk Fungal Biodiversity Institute
-	DTO 188-A9	wild type	-	Cinnamon, imported to the Netherlands	*Aspergillus niger*	Westerdijk Fungal Biodiversity Institute
-	DTO 203-H4	wild type	-	Soil, Kabodan island, Iran	*Aspergillus niger*	Westerdijk Fungal Biodiversity Institute
-	DTO 225-H3	wild type	-	Raisins, imported to Denmark	*Aspergillus niger*	Westerdijk Fungal Biodiversity Institute
-	DTO 240-E2	wild type	-	Breakfast cereal, Turkey	*Aspergillus niger*	Westerdijk Fungal Biodiversity Institute
-	DTO 240-H6	wild type	-	Muesli, Turkey	*Aspergillus niger*	Westerdijk Fungal Biodiversity Institute
-	DTO 240-I6	wild type	-	Dried fig, Turkey	*Aspergillus niger*	Westerdijk Fungal Biodiversity Institute
-	DTO 240-I9	wild type	-	Dried fruit, Turkey	*Aspergillus niger*	Westerdijk Fungal Biodiversity Institute
-	DTO 241-B2	wild type	-	Breakfast cereal, Turkey	*Aspergillus niger*	Westerdijk Fungal Biodiversity Institute
-	DTO 241-B7	wild type	-	Muesli, Turkey	*Aspergillus niger*	Westerdijk Fungal Biodiversity Institute
-	DTO 257-G2	wild type	-	Filling, the Netherlands	*Aspergillus niger*	Westerdijk Fungal Biodiversity Institute
-	DTO 260-C2	wild type	-	Indoor, school, Turkey	*Aspergillus niger*	Westerdijk Fungal Biodiversity Institute
-	DTO 267-I2	wild type	-	House dust, Thailand	*Aspergillus niger*	Westerdijk Fungal Biodiversity Institute
-	DTO 276-G2	wild type	-	BAL, Iran	*Aspergillus niger*	Westerdijk Fungal Biodiversity Institute
CBS 147343	DTO 291-B7	wild type	-	Coffee bean, Thailand	*Aspergillus niger*	Westerdijk Fungal Biodiversity Institute
-	DTO 293-E2	wild type	-	Coffee beans (Arabica), Thailand	*Aspergillus niger*	Westerdijk Fungal Biodiversity Institute
CBS 147344	DTO 293-G7	wild type	-	Coffee beans (Robusta), Thailand	*Aspergillus niger*	Westerdijk Fungal Biodiversity Institute
CBS 133816	DTO 316-E3	wild type	-	Black pepper, Denmark	*Aspergillus niger*	Westerdijk Fungal Biodiversity Institute
CBS 147345	DTO 316-E4	wild type	-	USA	*Aspergillus niger*	Westerdijk Fungal Biodiversity Institute
-	DTO 316-E5	wild type	-	Raisins, California, USA	*Aspergillus niger*	Westerdijk Fungal Biodiversity Institute
-	DTO 316-E6	wild type	-	Raisins, California, USA	*Aspergillus niger*	Westerdijk Fungal Biodiversity Institute
CBS 147346	DTO 321-E6	wild type	-	CF patient material, the Netherlands	*Aspergillus niger*	Westerdijk Fungal Biodiversity Institute
CBS 147347	DTO 326-A7	wild type	-	Petri dish in soft drink factory, the Netherlands	*Aspergillus niger*	Westerdijk Fungal Biodiversity Institute
-	DTO 337-D3	wild type	-	Fruit, Belgium	*Aspergillus niger*	Westerdijk Fungal Biodiversity Institute
-	DTO 355-F9	wild type	-	Patient material, the Netherlands	*Aspergillus niger*	Westerdijk Fungal Biodiversity Institute
-	DTO 360-C1	wild type	-	Liquorice solution, the Netherlands	*Aspergillus niger*	Westerdijk Fungal Biodiversity Institute
CBS 115.50	DTO 367-B6	wild type	-	Unknown	*Aspergillus niger*	Westerdijk Fungal Biodiversity Institute
CBS 281.95	DTO 367-C9	wild type	-	Unknown	*Aspergillus niger*	Westerdijk Fungal Biodiversity Institute
CBS 769.97	DTO 367-D1	wild type	-	Leather	*Aspergillus niger*	Westerdijk Fungal Biodiversity Institute
CBS 115989	DTO 367-D6	wild type	-	Unknown	*Aspergillus niger*	Westerdijk Fungal Biodiversity Institute
CBS 116681	DTO 367-D7	wild type	-	Imported kernels of apricots, the Netherlands	*Aspergillus niger*	Westerdijk Fungal Biodiversity Institute
CBS 119394	DTO 367-E2	wild type	-	USA	*Aspergillus niger*	Westerdijk Fungal Biodiversity Institute
CBS 121997	DTO 367-E9	wild type	-	Coffee bean, Chiangmai, Thailand	*Aspergillus niger*	Westerdijk Fungal Biodiversity Institute
CBS 129379	DTO 367-G3	wild type	-	Soil, Cedrus deodar forest, Mussoorie, India	*Aspergillus niger*	Westerdijk Fungal Biodiversity Institute
CBS 132413	DTO 367-G7	wild type	-	Soil, 200m from W. mirabilis, Swakop, Namibia	*Aspergillus niger*	Westerdijk Fungal Biodiversity Institute
CBS 133817	DTO 367-G8	wild type	-	Black pepper, Denmark	*Aspergillus niger*	Westerdijk Fungal Biodiversity Institute
CBS 133818	DTO 367-G9	wild type	-	Raisins, Denmark	*Aspergillus niger*	Westerdijk Fungal Biodiversity Institute
CBS 140837	DTO 367-H2	wild type	-	Soil, Rudňany, Slovakia	*Aspergillus niger*	Westerdijk Fungal Biodiversity Institute
-	DTO 368-H7	wild type	-	K-sorbate free margarine, the Netherlands	*Aspergillus niger*	Westerdijk Fungal Biodiversity Institute
-	DTO 368-H8	wild type	-	Beverages factory, India	*Aspergillus niger*	Westerdijk Fungal Biodiversity Institute
-	DTO 368-H9	wild type	-	Ice tea red, Philippines	*Aspergillus niger*	Westerdijk Fungal Biodiversity Institute
CBS 147352	DTO 368-I1	wild type	-	Air next to bottle blower, Mexico	*Aspergillus niger*	Westerdijk Fungal Biodiversity Institute
-	DTO 368-I2	wild type	-	Decaffinated tea bags, Belgium	*Aspergillus niger*	Westerdijk Fungal Biodiversity Institute
-	DTO 368-I3	wild type	-	Environment in factory, Uzbekistan	*Aspergillus niger*	Westerdijk Fungal Biodiversity Institute
-	DTO 368-I4	wild type	-	Potassium sorbate containing margarine, Ghana	*Aspergillus niger*	Westerdijk Fungal Biodiversity Institute
-	DTO 368-I5	wild type	-	Food factory of Sanquinetto, Italy	*Aspergillus niger*	Westerdijk Fungal Biodiversity Institute
CBS 147353	DTO 368-I6	wild type	-	Food factory of Sanquinetto, Italy	*Aspergillus niger*	Westerdijk Fungal Biodiversity Institute
-	DTO 368-I7	wild type	-	Used in soy sauce fermentation process, China	*Aspergillus niger*	Westerdijk Fungal Biodiversity Institute
CBS 554.65	DTO 368-I8	wild type	-	Connecticut, USA	*Aspergillus niger*	Westerdijk Fungal Biodiversity Institute
MA985.18		complementation *sdrA* G1296A	CBS 147320	-	*Aspergillus niger*	This study
MA985.34		complementation *sdrA* G1296A	CBS 147320	-	*Aspergillus niger*	This study
MA985.52		complementation *sdrA* G1296A	CBS 147320	-	*Aspergillus niger*	This study
MA985.54		complementation *sdrA* G1296A	CBS 147320	-	*Aspergillus niger*	This study
MA234.1		∆*kusA*	N402	-	*Aspergillus niger*	[34]
SJS148.1		Δ*warA*	MA234.1	-	*Aspergillus niger*	This study
SJS157.1		Δ*sdrA*	MA234.1	-	*Aspergillus niger*	This study
SJS158.1		Δ*warB*	MA234.2	-	*Aspergillus niger*	This study
SJS159.1		Δ*sdrA,* Δ*warA*	SJS148.1	-	*Aspergillus niger*	This study
SJS160.2		Δ*warA,* Δ*warB*	SJS148.2	-	*Aspergillus niger*	This study
SJS161.1		Δ*sdrA,* Δ*warB*	MA234.1	-	*Aspergillus niger*	This study
SJS162.1		Δ*sdrA,* Δ*warA,* Δ*warB*	SJS148.1	-	*Aspergillus niger*	This study

**Table 2 microorganisms-10-00221-t002:** Primers used in this study.

Primer Name	Sequence	Function
p1f sjs28	TCCCGCATCGGCTAAGTCTCCA	*sdrA* repair DNA 1 for CBS 147320
p2r sjs28	CTGATTCCGCTTCATTCGCAGCACGCGGTCAATCTCT	*sdrA* repair DNA 1 for CBS 147320
p3f sjs28	GAATGAAGCGGAATCAGCGCGAGGCTCGAGCGTGTTA	*sdrA* repair DNA 2 for CBS 147320
p5r sjs28	GGTCACGCAGATATGGCTGAG	*sdrA* repair DNA 2 for CBS 147320
TS1_sdrA_fw	TCCCGCATCGGCTAAGTCTCCA	Creation of 5′ *sdrA* flank, 367 bp
TS1_sdrA_rv	GGAGTGGTACCAATATAAGCCGGCGGTGTGTCGGAACCTCAAAAGC	Creation of 5′ *sdrA* flank, 367 bp
TS2_sdrA_fw	CCGGCTTATATTGGTACCACTCCCCATGACGTTATGCGGCCCCTC	Creation of 3′ *sdrA* flank, 502 bp
TS2_sdrA_rv	AGTGGCACCCGTCATGGCTACT	Creation of 3′ *sdrA* flank, 502 bp
sdrA_sgRNA2_fw	AATGAAACGCAATCAGCGCGGTTTTAGAGCTAGAAAT	Create the *sdrA* target for the CRISPR/Cas9 plasmid
sdrA_sgRNA2_rv	CGCGCTGATTGCGTTTCATTGACGAGCTTACTCGTTT	Create the *sdrA* target for the CRISPR/Cas9 plasmid
diag_sdrA_fw	ACTTAGGGGGTGGGACCAGTGG	Diagnostic PCR *sdrA* deletion
diag_sdrA_rv	GGACTTTGATGCCGAGCATGGC	Diagnostic PCR *sdrA* deletion
5_warA_fw	GGCGTCCTCCAGGGTCTCATCT	Creation of 5′ *warA* flank, 368 bp
5_warA_rv	GGAGTGGTACCAATATAAGCCGGTGGCTTGCTGTTATTCTAGAGAGGG	Creation of 5′ *warA* flank, 368 bp
3_warA_fw	CCGGCTTATATTGGTACCACTCCTGTGTATTTGTCTGGAGTGGATGT	Creation of 3′ *warA* flank, 1002 bp
3_warA_rv	AGCTCCCGCTCAATCCTCGAGA	Creation of 3′ *warA* flank, 1002 bp
warA_sgRNA_fw	CGATAGACGATGCTTACCTGGTTTTAGAGCTAGAAAT	Create the *warA* target for the CRISPR/Cas9 plasmid
warA_sgRNA_rv	CAGGTAAGCATCGTCTATCGGACGAGCTTACTCGTTT	Create the *warA* target for the CRISPR/Cas9 plasmid
diag_warA_fw	CACAATGCCATGTAGCGCGCAA	Diagnostic PCR *warA* deletion
diag_warA_rv	ACACGATCTGACCGCGATGACG	Diagnostic PCR *warA* deletion
TS1_warB_fw	TCGACCCTCCCGGTTTGGTCAA	Creation of 5′ *warB* flank, 599 bp
TS1_warB_rv	GGAGTGGTACCAATATAAGCCGGTGAAGGAGGTTTGGTTGCGGGT	Creation of 5′ *warB* flank, 599 bp
TS2_warB_fw	CCGGCTTATATTGGTACCACTCCACGATACGACGAAGTTCAGCAT	Creation of 3′ *warB* flank, 544 bp
TS2_warB_rv	AGTTCGGCCACTTCTCGGACCA	Creation of 3′ *warB* flank, 544 bp
warB_sgRNA2_rv	CGGTGTTCTCTTCGAAGCGCGACGAGCTTACTCGTTT	Create the *warB* target for the CRISPR/Cas9 plasmid
warB_sgRNA2_fw	GCGCTTCGAAGAGAACACCGGTTTTAGAGCTAGAAAT	Create the *warB* target for the CRISPR/Cas9 plasmid
diag_warB_fw	TCGCCCTCGTCTTACTCCTCCC	Diagnostic PCR *warB* deletion
diag_warB_rv	CCATGACGTCCTCCATCACCGC	Diagnostic PCR *warB* deletion

**Table 3 microorganisms-10-00221-t003:** All plasmids used in this study.

Plasmid Name	Target Sequence	Function	Origin
pTLL108.1	-	Template for the amplification of guide RNA	[37]
pTLL109.2	-	Template for the amplification of guide RNA	[37]
pFC332	-	Backbone containing CRISPR/Cas9	[38]
pMA433.2	AATGAAACGCAATCAGCGCG	Targeted double-stranded break in *sdrA* gene	This study
pMA434.1	CGATAGACGATGCTTACCTG	Targeted double-stranded break in *sdrA* gene	This study
pMA435.2	GCGCTTCGAAGAGAACACCG	Targeted double-stranded break in *sdrA* gene	This study

**Table 4 microorganisms-10-00221-t004:** The average and most extreme sorbic MIC_u_ values (average ± SD) out of 100 *A. niger* strains grown in MEB and commercial ice tea.

Sample	MIC_u_ in MEB (mM)		MIC_u_ in Ice Tea (mM)	
	7 days	28 days	7 days	28 days
Average of 100 strains	3.7 ± 0.6	4.8 ± 0.8	2.9 ± 0.4	3.8 ± 0.5
CBS 147320 (sorbic-acid-sensitive strain)	2.5 ± 0.7	4.0 ± 1.4	2.0 ± 0.0	3.0 ± 0.0
CBS 113.50 (sorbic-acid-resistant strain)	6.5 ± 0.7	7.0 ± 0.0	4.0 ± 0.0	5.5 ± 0.7

## Data Availability

Plasmids and strains are available upon request. Genome data on a subset of *A. niger* strains discussed in this manuscript are available at NCBI under BioProject ID PRJNA743902. All other data is included as part of the manuscript and Appendix A.

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
