# Peer review of "Natural Variation and the Role of Zn2Cys6 Transcription Factors SdrA, WarA and WarB in Sorbic Acid Resistance of Aspergillus niger"

_microorganisms, 2022, doi:10.3390/microorganisms10020221_

Round 1
Reviewer 1 Report
General comments.
This original study represents a major advance in our understanding of the sorbate resistance of A. niger. The results include a screen of sorbic acid resistance on 100 A. niger wild-type strains, which appears to show only a modest degree of variability in sorbic acid resistance in strain collections. Additionally, it also describes the screening of 240 transcription factor knock-out strains; the construction and screening of single, double and triple sdrA and warA and warB transcription factor knockout strains – the latter leading to the first demonstration that all three of these transcription factors, sdrA and warA and warB, contribute to sorbic acid resistance.
Specific comments.
l62-76 It is surprising to see the “classic weak acid theory” still being given such a strong emphasis in the text of a paper nowadays. Over the past 20 years many studies have shown that this is a vastly oversimplistic, outdated view of the effects of weak acids on fungal growth; effects that become increasingly more complex as the acid becomes more lipophilic. As regards sorbate, there have been several studies - only a few of which are cited here – showing that resistance involves, amongst other factors, modification of the acid, active extrusion of the acid, cell envelope structure, glycolysis inhibition (notably at the phosphofructokinase step), the need to maintain a reducing intracellular environment (pentose phosphate pathway), the need for an acidified vacuole and endocytosis. In addition, sorbate also causes severe oxidative stress, by elevating free radical production by the mitochondrial electron transport chain (causing MIC values for an organism such as Z. bailii to be very different -/+ oxygen). It is therefore not surprising that the sorbic acid stress response overlaps with other stress responses)
L82 In stating “Negatively influencing respiration” are we meaning ATP generated by respiration (greatly decreased by sorbate) or uncoupled mitochondrial electron transport (increased by sorbate)??
L254 A premature stop is a “nonsense” mutation; a “missense” mutation is where the codon change causes incorporation of an alternative aminoacid.
L378-393 It is interesting that the screen of transcription factor deletants identified sensitivity with loss of AtfA, HapX, NsdC and AcuK. This would appear to confirm indications from the screens of the yeast deletant collection that sorbic acid resistance requires maintenance of a reducing intracellular environment. This in turn will allow iron uptake and mitochondrial Fe-S cluster synthesis.
Reviewer 2 Report
The study investigated three novel transcription factors related to sorbic acid resistence of Aspergillus Níger. In general, presented study design and data are of acceptable quality. I only have a few minor comments for consideration
- Figure S1. Why was a target site more than 1000 bps away from the stop codon selected? Conventionally, a PAM site as close to the stop codon as possible would be preferred? Readers would appreciate some explainations regarding the authors' approach?
- What's gRNA efficiency? The data would be necessary to support the CRISPR design?
- Was off-target effect evaluated? If so the data ought to be provided.
